

# Comparison of miRNA expression profiles in pituitary–adrenal axis between Beagle and Chinese Field dogs after chronic stress exposure

Wei Luo[1,2,*], Meixia Fang[3,*], Haiping Xu[1,2], Huijie Xing[3], Jiangnan Fu[3] and Qinghua Nie[1,2]

[1] Department of Animal Genetics, Breeding and Reproduction, College of Animal Science, South China Agricultural University, Guangzhou, Guangdong, China
[2] Guangdong Provincial Key Lab of Agro-Animal Genomics and Molecular Breeding and National-Local Joint Engineering Research Center for Livestock Breeding, South China Agricultural University & Guangdong Wens Food Corporation, Guangzhou, Guangdong, China
[3] Institute of Laboratory Animals, Jinan University, Guangzhou, Guangdong, China
[*] These authors contributed equally to this work.

Corresponding authors
Jiangnan Fu, fujiangnan126@126.com
Qinghua Nie, nqinghua@scau.edu.cn

## ABSTRACT

MicoRNAs (miRNAs), usually as gene regulators, participate in various biological processes, including stress responses. The hypothalamus–pituitary–adrenal axis (HPA axis) is an important pathway in regulating stress response. Although the mechanism that HPA axis regulates stress response has been basically revealed, the knowledge that miRNAs regulate stress response within HPA axis, still remains poor. The object of this study was to investigate the miRNAs in the pituitary and adrenal cortex that regulate chronic stress response with high-throughput sequencing. The pituitary and adrenal cortex of beagles and Chinese Field dogs (CFD) from a stress exposure group (including beagle pituitary 1 (BP1), CFD pituitary 1 (CFDP1), beagle adrenal cortex 1 (BAC1), CFD adrenal cortex 1 (CFDAC1)) and a control group (including beagle pituitary 2 (BP2), CFD pituitary 2 (CFDP2), beagle adrenal cortex 2 (BAC2), CFD adrenal cortex 2 (CFDAC2)), were selected for miRNA-seq comparisons. Comparisons, that were made in pituitary (including BP1 vs. BP2, CFDP1 vs. CFDP2, BP1 vs. CFDP1 and BP2 vs. CFDP2) and adrenal cortex (including BAC1 vs. BAC2, CFDAC1 vs. CFDAC2, BAC1 vs. CFDAC1 and BAC2 vs. CFDAC2), showed that a total of 39 and 18 common differentially expressed miRNAs (DE-miRNAs) (Total read counts > 1,000, Fold change > 2 & $p$-value < 0.001), that shared in at least two pituitary comparisons and at least two adrenal cortex comparisons, were detected separately. These identified DE-miRNAs were predicted for target genes, thus resulting in 3,959 and 4,010 target genes in pituitary and adrenal cortex, respectively. Further, 105 and 10 differentially expressed genes (DEGs) (Fold change > 2 & $p$-value < 0.05) from those target genes in pituitary and adrenal cortex were obtained separately, in combination with our previous corresponding transcriptome study. Meanwhile, in line with that miRNAs usually negatively regulated their target genes and the dual luciferase reporter assay, we finally identified cfa-miR-205 might play an important role by upregulating *MMD* in pituitary and hippocampus, thus enhancing the immune response, under chronic stress exposure. Our results shed light on the miRNA expression profiles in the pituitary and adrenal cortex with and without chronic stress exposure, and provide a new insight

into miR-205 with its feasible role in regulating chronic stress in the pituitary and hippocampus through targeting *MMD*.

## INTRODUCTION

MiRNAs belong to a class of non-coding endogenous RNAs with the lengths ranging from 18 nt to 25 nt. They have been much explored as gene expression regulators that involve many biological processes, including differentiation, proliferation, development and apoptosis of cells, hormone secretions, virus diseases and cancers (*Stefani & Slark, 2008*; *Cordes, Srivastava & Ivey, 2010*; *Taft et al., 2010*). MiRNAs are also important in the neuroendocrine system and functionally associated with postembryonic development, axon guidance, synaptic plasticity and astrocyte activity (*Bartel, 2004*; *Krichevsky et al., 2006*; *Mor et al., 2011*). Previous studies showed miR-18a could downregulate the expression of glucocorticoid receptor *in vitro* (*Uchida et al., 2008*; *Vreugdenhil et al., 2009*). And in paraventricular nucleus, the miR-18a expression levels of the excessively stress-induced rats were higher than that in the natural rats. In addition, with the neuron treated by glucocorticoid (*Kawashima et al., 2010*), the miRNA-132 in neuron showed a lower expression level compared with that in negative control. Experiments in rats indicated that, compared with non-stress groups, the expression levels of some miRNAs in prefrontal cortex changed a bit within the acute stress exposure groups, but significantly upregulated within the chronic stress exposure groups (*Rinaldi et al., 2010*). Collectively, the existing knowledge indicates that the miRNAs play important roles in the neuroendocrine system and are closely associated with the stress response.

The hypothalamus–pituitary–adrenal axis (HPA axis) is an important part of the neuroendocrine system. Stress response, including acute and chronic ones, is regulated by the activity of the HPA axis in part (*Frodl & O'Keane, 2013*; *Griffiths & Hunter, 2014*). HPA axis is involved in maintaining the homeostasis of the body, with its regulations in digestion, immune system, mood, sex behavior, energy storage and consumption (*Bodera, Stankiewicz & Kocik, 2014*; *Kennedy et al., 2014*; *Gelman et al., 2015*). Currently, the mechanism of HPA axis in regulating stress response has been basically revealed. However, especially in dogs, the knowledge of the miRNAs associated with stress response in HPA axis is still devoid of a systemic revelation. In addition, to our knowledge, the systemic study of miRNAs in dog tissues still remains poor compared with that in other frequently-used medical animals, just with a study concerning dog tracheas and dog lungs reported (*Zhao et al., 2014*). In 2008, 357 candidate miRNAs in the Canis familiaris genome were identified with a comparative analysis of the whole genome in silico. Of them, 300 miRNAs were homological with characterized human miRNAs (*Zhou et al., 2008*). Virtually, miRNA is characterized by its high homology among species, thus the miRNA study targeting HPA axis in dogs can further its corresponding understanding of that in human or other animals.

High-throughput sequencing has been a powerful method to investigate the expression profiles of miRNAs, and has been widely utilized in various organisms (*Zhao et al., 2014*; *Zhan et al., 2014*; *Li et al., 2015a*; *Wongwarangkana et al., 2015*). In this study, we aimed to investigate the miRNAs involving chronic stress response in dog pituitary and dog adrenal cortex (pituitary–adrenal axis). To achieve it, the pituitary and adrenal cortex tissues of beagles and Chinese Field dogs (CFD) that were treated with chronic stress exposure and non-treatment, were separately performed with miRNA-seq. Then, in pituitary and adrenal cortex, miRNA profiles were compared between the chronic stress exposure groups and the control groups within and between breeds to identify the miRNAs associated with chronic stress response in the pituitary–adrenal axis.

To better pinpoint the pivotal miRNAs of regulating chronic stress in pituitary and adrenal cortex, we combined the miRNA-seq of this study with our previous transcriptome study (*Luo et al., 2015*). Importantly, the animals used, stress exposure treatment, sampling and differential expression analysis strategy in this study were all the same with our previous transcriptome study (*Luo et al., 2015*). In our transcriptome study, glucocorticoid levels of pre- and post-stress exposure in each day were detected, and hippocampal sections of stress exposure groups and control groups in both breeds were conducted, both giving evidence that our stress exposure was potent and valid. In addition, our previous transcriptome study identified a total of 511 and 171 differentially expressed genes in pituitary and adrenal cortex, respectively (Table S1).

The two dog breeds used here, including Beagle and CFD, have distinct characteristics in stress tolerance. Beagles are manageable, have good environmental adaptability and good stress tolerance, while CFD is exciteable. We herein utilize two breeds that are distinct in stress tolerance but not just a single breed to avoid drawing a sweeping conclusion and pinpoint the miRNAs concerning chronic stress response more accurately.

## MATERIAL AND METHOD

### Ethics statements

In this experiment, all animals used were approved by the Animal Care Committee of Jinan University (Guangzhou, People's Republic of China) approval number 20131018001, and strictly implemented in line with the experimental basic principles.

### Chronic stress treatment

Chronic stress treatment was given briefly as follows: 6 unrelated purebred male Chinese Field dogs (CFD) and 6 unrelated purebred male beagles, similar in health, weight, and other aspects were selected randomly. From each of these two breeds, three dogs were selected for stress exposure via intermittent electrical stimulation and restraint stress, and the other 3 dogs of each breed, used as controls, were not exposed to the stress. Each morning for 10 days, dogs were restrained and electrically stimulated with a stable current of 10 mA for 6 s followed by a 6-s interval without stimulation, lasting for 20 min every day. Meanwhile, before and after each 20-min stress exposure session, 4 ml of blood was collected and isolated to examine cortisol level with a cortisol radiation immunoassay kit used.

## Samples and RNA preparation

All 12 dogs were killed by air embolism on the 11th day, along with all parts of the adrenal cortex and pituitary tissues collected and fast frozen in liquid nitrogen until for miRNA-seq. Further, Trizol reagent (Invitrogen, USA) was used to isolate the RNAs of all the adrenal cortex and pituitary tissues collected above. The quantity and quality of RNA were examined by Agarose gel electrophoresis, NanoDrop 2000 (Thermo, USA), and Agilent 2100 (Agilent, USA). The RNA collected from each group, including 3 dogs of each, were pooled with equal mass respectively, thus obtaining 8 RNA pooled samples, including CFDP1 (CFD pituitary with stress exposure), CFDP2 (CFD pituitary with non-disposal), CFDAC1 (CFD adrenal cortex with stress exposure), CFDAC2 (CFD adrenal cortex with non-disposal), BP1 (beagle pituitary with stress exposure), BP2 (beagle pituitary with non-disposal), BAC1 (beagle adrenal cortex with stress exposure), and BAC2 (beagle adrenal cortex with non-disposal). In addition, tissues from the hippocampus were collected to perform slice analysis (Nissl staining), aiming to evaluate the influences of stress exposure on hippocampal region cells.

## Small RNA library construction and sequencing

A total of 1 μg RNA was collected from each RNA pooled sample above. Then, specific 3′ and 5′ adaptors were added to the RNA ends of each group (Truseq TM Small RNA sample prep Kit, Illumina). After that, the RNA with adaptors were reverse transcribed into its 1st cDNA, with random primers added. PCR was performed to amplify the cDNA with 11–12 cycles run and then resulted in 8 cDNA libraries, including CFDP1, CFDP2, CFDAC1, CFDAC2, BP1, BP2, BAC1 and BAC2. After purified further by polyacrylamide gel electrophoresis (PAGE), each cDNA library was performed by 1*50 bp high-sequencing on Hiseq-2000 (Majorbio Inc, Shanghai, China).

## Sequence data analysis

The raw reads obtained by high-sequencing were further performed with quality control to result in clean reads. In brief, the quality control was that of trimming low-quality reads (ambiguous N and length < 18 nt), 3′ adapters, 5′ adapters and poly (A) sequences, with SeqPrep (https://github.com/jstjohn/SeqPrep) and Sickle (https://github.com/najoshi/sickle) used. Thereby, the lengths of 18–32 bp reads, as the clean ones, were obtained and calculated for their length distributions, using Fastx-Toolkit (http://hannonlab.cshl.edu/fastx_toolkit/). Meanwhile, the identical reads were collapsed to obtain the unique reads for the statistics of the small RNA's species and abundance. Moreover, the clean reads were mapped to Rfam database (ftp://sanger.ac.uk/pub/databases/Rfam/) and GenBank noncoding RNA database (http://blast.ncbi.nlm.nih.gov/) to annotate the miscellaneous RNAs, thus filtering out the rRNA, scRNA, snoRNA, snRNA, tRNA and other non-coding RNA. The remaining sRNAs obtained above, including known miRNAs and unknown miRNAs, were then mapped to the CanFam 3.1 genome to analyze their distributions on genome and calculate their expressive quantity. In addition, the remaining sRNAs were mapped to the Canis Canine miRNA data of miRBase 21.0 to identify the known miRNAs. Meanwhile, the novel miRNAs were predicted
using miRDeep2 (http://www.mdcberlin.de/en/research/research_teams/systems_biology_ of_gene_regulatory_elements/projects/miRDeep/index.html) from unannotated sRNAs (*Bonnet et al., 2004*; *Langmead et al., 2009*; *Friedländer et al., 2012*). The data discussed here have been deposited in NCBI Sequence Read Archive and is accessible through GEO Series accession no. GSE72015 (http://www.ncbi.nlm.nih.gov/geo/query/acc.cgi?acc=GSE72015).

## Analysis of differentially expressed miRNAs

To explore the differentially expressed miRNAs (DE-miRNAs) in pituitary and adrenal, on the basis of miRNA expression profiles, eight comparisons were made, including BP1_vs_BP2, CFDP1_vs_CFDP2, BP1_vs_CFDP1 and BP2_vs_CFDP2 in pituitary; BAC1_vs_BAC2, CFDAC1_vs_CFDAC2, BAC1_vs_CFDAC1 and BAC2_vs_CFDAC2 in adrenal cortex. Here, R package DEGseq using a Benjamini $q$-value of 0.001 as the cut-off (corrected $p$-value < 0.001) (http://www.bioconductor.org/packages/release/bioc/html/ DEGseq.html) (*Wang et al., 2010*), combined with Value1/Value2 > 2 (Value indicated the normalized count of samples), were used to screen out the DE-miRNAs. Furthermore, the common miRNAs that differentially expressed in at least two comparisons from pituitary and adrenal cortex were selected for the following target prediction, respectively.

## Target gene prediction of DE-miRNA and its GO, KEGG analysis

For the target gene prediction, we utilized Targetscan 7.0 (http://www.targetscan.org/vert_ 70/). Importantly, target genes predicted with the cumulative weighted context ++ score less than −0.2, were selected for further analysis. Then, the target genes were selected to perform GO and KEGG pathway analysis, using the KOBAS 2.0 Functional Annotation Tool (http://kobas.cbi.pku.edu.cn/program.inputForm.do?program=Annotate) (Corrected $p$-value < 0.05).

## Conjoint analysis of DE-miRNAs and DEGs

To better identify the important miRNAs in regulating chronic stress response, the DEGs identified in our previous transcriptome study of the pituitary and adrenal cortex were compared with the target genes of DE-miRNAs in the pituitary and adrenal cortex separately, thereby obtaining the genes that were both DEGs and target genes in pituitary and adrenal cortex, respectively (as a matter of convenience, the genes that were both DEGs and target genes were named "DE-target genes"). Further, according to a negative correlation between miRNAs and their target genes in expression pattern, the DE-miRNAs and their corresponding DE-target genes that went against this pattern, were then removed, thus resulting in the final DE-miRNA and its corresponding DE-target gene (named DE-(miRNA-mRNA) pairs hereafter). Subsequently, combined with the expression level of DE-miRNAs and their corresponding DE-target genes' relatedness to stress regulation, a number of DE-(miRNA-mRNA) pairs were selected to perform dual luciferse reporter assay.

## Validation of DE-miRNAs by qPCR

To elucidate the validity of the miRNA-seq data, DE-miRNAs, including miR-30a, miR-124, miR-205 and miR-222, were further detected by qPCR. In each group, 1 µg of pooled RNA

**Table 1** Statistics of miRNA-seq across 8 samples in brief.

| | BAC2 | BAC1 | CFDAC2 | CFDAC1 | BP2 | BP1 | CFDP2 | CFDP1 |
|---|---|---|---|---|---|---|---|---|
| Raw reads number | 28,667,214 | 15,926,611 | 32,583,375 | 32,665,593 | 27,591,701 | 18,359,529 | 81,028,035 | 51,202,889 |
| Clean reads number | 24,078,122 | 11,900,933 | 27,143,726 | 25,448,102 | 24,957,693 | 16,409,592 | 74,445,014 | 40,249,802 |
| Unique reads | 975,309 | 693,607 | 1,508,439 | 1,118,474 | 646,032 | 477,154 | 1,644,704 | 1,491,566 |
| Perfectly matched unique reads | 211,901 | 178,920 | 84,239 | 257,486 | 175,320 | 140,709 | 426,965 | 423,546 |
| Unique gene miRNA number | 68,246 | 48,170 | 100,880 | 68,735 | 68,482 | 56,260 | 167,784 | 89,791 |
| Total percent of miRNA in clean reads | 82.08% | 73.63% | 74.32% | 69.32% | 88.97% | 88.42% | 88.27% | 70.16% |

**Notes.**
BAC1, beagle adrenal cortex with treatment; BAC2, beagle adrenal cortex without treatment; BP1, beagle pituitary with treatment; BP2, beagle pituitary without treatment; CFDAC1, Chinese Field dog adrenal cortex with treatment; CFDAC2, Chinese Field dog adrenal cortex without treatment; CFDP1, Chinese Field dog pituitary with treatment; CFDP2, Chinese Field dog pituitary without treatment.

was reverse transcribed using ReverTra Ace qPCR RT Kit 101 (TOYOBO, Japan), along with bulge-loop RT primers added. The bulge-loop RT primers and primers for qPCR were both synthesized by RIBOBIO (Guangzhou, China). QPCR was performed on the Bio-Rad S1000 with Bestar SYBR Green RT-PCR Master Mix (DBI Bioscience, Germany). Besides, *U6* and *U48*, as the reference genes, were used to normalize the miRNA expression on the basis of $2^{-\Delta\Delta CT}$ method (*Schmittgen & Livak, 2008*). The detailed information of the primers and the annealing temperatures was presented in Table S2.

## RESULTS

### High-sequencing of small RNA

In this study, eight libraries including BAC2, BAC1, CFDAC2, CFDAC1, BP2, BP1, CFDP2 and CFDP1 were constructed for miRNA-seq. As a result, a number of raw reads ranging from 15,926,611 to 81,028,035 were obtained from these 8 groups (Table 1). After eliminating reads of low quality (ambiguous N and length < 18 nt), 3′ adapters, 5′ adapters and poly (A) sequences, a number of clean reads ranging from 11,900,933 to 74,445,014 were obtained from the 8 groups (Table 1). Further, the clean reads from each group were mapped to Rfam database (11.0, http://Rfam.sanger.ac.uk/) and GenBank noncoding RNA database (http://blast.ncbi.nlm.nih.gov/) to annotate miscellaneous RNAs. And the detailed annotations of small RNA in each library were presented in Fig. S1.

The statistics of size distributions showed that the majority of sRNAs were 21–23 nt in length across 8 samples, along with a significantly downregulated profile in post-stress group compared with that in pre-stress (Figs. 1A and 1B). Besides, when mapped to the reference genome of Canis Canine with the remaining sRNA reads (including known miRNAs and unknown miRNAs), Chromosomes 1, 3, 5, 20 and 31 were found to be mapped most abundantly (ratio > 5%) in dog (Fig. 2). Meanwhile, a bigger number of small RNA reads were showed in pituitary samples (especially in CFD pituitary samples) compared with that in adrenal cortex samples.

### Identification of known miRNAs

When mapped to the Canis Canine data of miRBase 21.0 by the remaining sRNA reads, a total of 425 out of 453 known miRNAs (Table S3) were identified across eight samples. Of
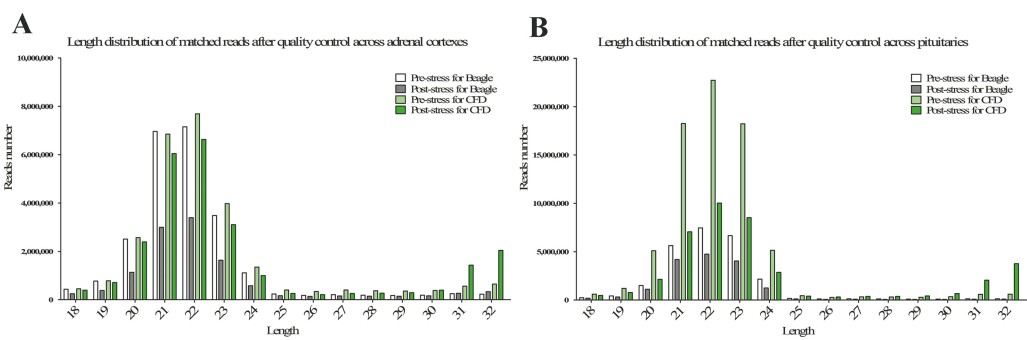

**Figure 1  Length distribution of matched reads after quality control in adrenal cortex (A) and pituitary (B).** The white, light black, light green, and green columns represent the samples of pre-stress for beagle, post-stress for beagle, pre-stress for CFD, and post-stress for CFD, respectively.

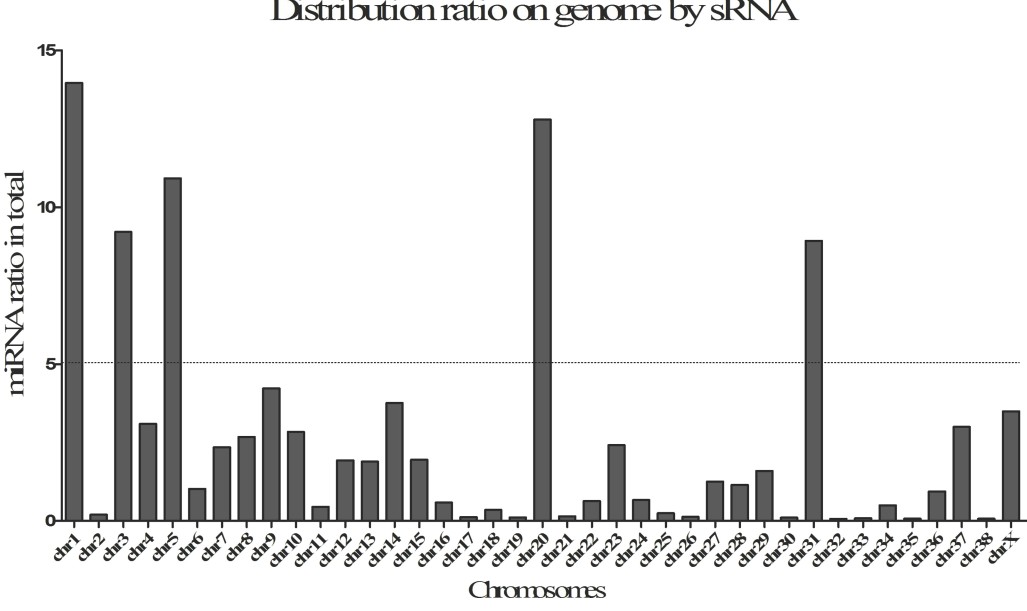

**Figure 2  Distribution ratio on by small RNAs (sRNA) on genome.**

the 425 identified known miRNAs, BAC2, BAC1, CFDAC2, CFDAC1, BP2, BP1, CFDP2 and CFDP1 contributed to 332, 325, 332, 345, 327, 318, 352 and 355 miRNAs, respectively.

Besides, the importance attached to the abundance of the miRNA expression in these two breeds was conducive to identify the miRNAs of high activity in dog pituitary and dog adrenal cortex. In adrenal cortex samples, cfa-miR-99a was the most abundant known miRNA, followed with cfa-miR-30 family (including cfa-miR-30a, cfa-miR-30b, cfa-miR-30c and cfa-miR-30d), cfa-miR-26 family (including cfa-miR-26a and cfa-miR-26b) and cfa-miR-7 family (including cfa-miR-7 and cfa-miR-7g) (The top 20 known miRNAs of abundance in adrenal cortex were listed in Table S4). Meanwhile, cfa-miR-7 family (including cfa-miR-7 and cfa-miR-7g), cfa-miR-99a, cfa-miR-30 family (including cfa-miR-30a and cfa-miR-30d), and cfa-miR-125 family (including cfa-miR-125a and

**Table 2** **39 and 18 common differentially expressed miRNAs pituitary from the pituitary and adrenal cortex respectively.** MiRNAs ranked up to down by the total reads number within adrenal cortex samples, where the total reads were greater than 1,000.

| miRNA ID | Normalized reads | | \|log2(Fold_change)\| |
|---|---|---|---|
| | Sample 1 | Sample 2 | |
| cfa-miR-30a | CFDP1:2553208 | CFDP2:5289957 | 1.05 |
| | BP1:6285675 | CFDP1:2553208 | 1.3 |
| cfa-miR-9 | CFDP1:2270770 | CFDP2:548039 | 2.05 |
| | BP1:788518 | CFDP1:2270770 | 1.53 |
| cfa-miR-124 | CFDP1:1806445 | CFDP2:67009 | 4.75 |
| | BP1:51801 | CFDP1:1806445 | 5.12 |
| | BP2:26408 | TP2:67009 | 1.34 |
| cfa-miR-144 | CFDP1:162116 | CFDP2:341015 | 1.07 |
| | BP1:561345 | CFDP1:162116 | 1.79 |
| cfa-miR-128 | CFDP1:439481 | CFDP2:86512 | 2.34 |
| | BP1:105386 | TP1:439481 | 2.06 |
| cfa-miR-146b | BP1:109353 | BP2:51299 | 1.09 |
| | BP2:51299 | BP1:163222 | 1.67 |
| cfa-miR-222 | CFDP1:153482 | CFDP2:29889 | 2.36 |
| | BP1:17161 | CFDP1:153482 | 3.16 |
| cfa-miR-34c | BP1:53532 | CFDP1:6620 | 3.02 |
| | BP2:77871 | BP1:12722 | 2.61 |
| cfa-miR-221 | CFDP1:90710 | CFDP2:19204 | 2.24 |
| | BP1:10704 | CFDP1:90710 | 3.08 |
| cfa-miR-95 | BP1:23657 | CFDP1:47878 | 1.02 |
| | BP2:13824 | CFDP2:31697 | 1.2 |
| cfa-miR-205 | BP1:10730 | BP2:22724 | 1.08 |
| | CFDP1:12765 | CFDP2:55847 | 2.13 |
| | BP2:22724 | CFDP2:55847 | 1.3 |
| cfa-miR-138a | CFDP1:52944 | CFDP2:7943 | 2.74 |
| | BP1:10264 | CFDP1:52944 | 2.37 |
| cfa-miR-135b | CFDP1:10065 | CFDP2:21001 | 1.06 |
| | BP1:28943 | CFDP1:10065 | 1.52 |
| cfa-miR-155 | BP1:12914 | CFDP1:26499 | 1.04 |
| | BP2:7255 | CFDP2:14735 | 1.02 |
| cfa-miR-885 | CFDP1:22660 | CFDP2:9447 | 1.26 |
| | BP1:7043 | CFDP1:22660 | 1.69 |
| cfa-miR-150 | CFDP1:17024 | CFDP2:4522 | 1.91 |
| | BP1:3981 | CFDP1:17024 | 2.1 |
| cfa-miR-326 | CFDP1:13307 | CFDP2:4126 | 1.69 |
| | BP1:5618 | CFDP1:13307 | 1.24 |
| cfa-miR-105a | BP1:2223 | BP2:1031 | 1.11 |
| | CFDP1:16712 | CFDP2:3167 | 2.4 |
| | BP1:2223 | CFDP1:16712 | 2.91 |
| | BP2:1031 | CFDP2:3167 | 1.62 |

| miRNA ID | Normalized reads | | \|log2(Fold_change)\| |
|---|---|---|---|
| | Sample 1 | Sample 2 | |
| cfa-miR-330 | CFDP1:9455 | CFDP2:3793 | 1.32 |
| | BP1:3954 | CFDP1:9455 | 1.26 |
| cfa-miR-219-3p | BP1:1118 | BP2:425 | 1.4 |
| | CFDP1:10445 | CFDP2:2774 | 1.91 |
| | BP1:1118 | CFDP1:10445 | 3.22 |
| | BP2:425 | CFDP2:2774 | 2.71 |
| cfa-miR-138b | CFDP1:8716 | CFDP2:1414 | 2.62 |
| | BP1:1571 | CFDP1:8716 | 2.47 |
| cfa-miR-202 | CFDP1:7366 | CFDP2:1598 | 2.2 |
| | BP1:1398 | CFDP1:7366 | 2.4 |
| cfa-miR-449 | BP1:1917 | BP2:4810 | 1.33 |
| | BP2:4810 | CFDP2:1484 | 1.7 |
| cfa-miR-139 | BP1:1651 | BP2:615 | 1.42 |
| | CFDP1:4883 | CFDP2:1124 | 2.12 |
| | BP1:1651 | CFDP1:4883 | 1.56 |
| cfa-miR-448 | BP1:1904 | BP2:693 | 1.46 |
| | CFDP1:2645 | CFDP2:919 | 1.53 |
| cfa-miR-224 | BP1:626 | BP2:1378 | 1.14 |
| | BP1:626 | CFDP1:2231 | 1.83 |
| cfa-miR-215 | CFDP1:597 | CFDP2:1940 | 1.7 |
| | BP2:754 | CFDP2:1940 | 1.36 |
| cfa-miR-34b | CFDP1:115 | CFDP2:269 | 1.22 |
| | BP1:1331 | CFDP1:115 | 3.53 |
| | BP2:1907 | CFDP2:269 | 2.82 |
| cfa-miR-1 | CFDP1:1505 | CFDP2:687 | 1.13 |
| | BP1:439 | BP2:1505 | 1.78 |
| cfa-miR-1307 | BP1:306 | BP2:875 | 1.51 |
| | BP2:355 | CFDP2:1375 | 1.95 |
| cfa-miR-802 | BP1:812 | BP2:347 | 1.23 |
| | CFDP1:115 | CFDP2:1086 | 3.24 |
| | BP1:812 | CFDP1:115 | 2.82 |
| | BP2:347 | CFDP2:1086 | 1.65 |
| cfa-miR-346 | CFDP1:1329 | CFDP2:234 | 2.51 |
| | BP1:200 | CFDP1:1329 | 2.73 |
| cfa-miR-1343 | BP1:8219 | CFDP1:22031 | 1.42 |
| | CFDP1:22031 | CFDP2:10113 | 1.12 |
| cfa-miR-769 | BP1:2087 | CFDP1:4473 | 1.1 |
| | CFDP1:4473 | CFDP2:1855 | 1.27 |
| cfa-miR-8903 | BP1:243 | CFDP1:815 | 1.75 |
| | CFDP1:815 | CFDP2:264 | 1.62 |
| cfa-miR-8908a-3p | BP1:4530 | BP2:1269 | 1.84 |
| | BP2:1269 | CFDP2:2759 | 1.12 |

| miRNA ID | Normalized reads | | |log2(Fold_change)| |
|---|---|---|---|
| | Sample 1 | Sample 2 | |
| cfa-miR-8908a-5p | BP1:1375 | BP2:349 | 1.98 |
| | BP1:1375 | CFDP1:515 | 1.42 |
| cfa-miR-8908b | BP1:1165 | BP2:264 | 2.14 |
| | BP1:1165 | CFDP1:565 | 1.04 |
| cfa-miR-8908d | BP1:1650 | BP2:286 | 2.53 |
| | BP1:1650 | CFDP1:798 | 1.05 |
| | BP2:286 | CFDP2:943 | 1.72 |
| cfa-miR-34a | BAC2:60814 | CFDAC2:172140 | 1.5 |
| | BAC1:84320 | CFDAC1:197301 | 1.23 |
| cfa-miR-338 | BAC1:62453 | BAC2:168721 | 1.43 |
| | BAC2:168721 | CFDAC2:79629.01 | 1.08 |
| cfa-miR-758 | BAC2:78102 | CFDAC2:33774.31 | 1.21 |
| | BAC1:61499 | CFDAC1:25712 | 1.26 |
| cfa-miR-9 | BAC1:24395 | BAC2:68265.21 | 1.48 |
| | BAC2:68265 | CFDAC2:27267.15 | 1.33 |
| cfa-miR-138a | BAC1:10468 | BAC2:35790 | 1.77 |
| | BAC2:35790 | CFDAC2:7959 | 2.17 |
| cfa-miR-124 | BAC1:6200 | BAC2:24560 | 1.99 |
| | BAC2:24560 | CFDAC2:5837 | 2.07 |
| cfa-miR-122 | BAC1:13068 | BAC2:591 | 4.47 |
| | CFDAC1:16590 | CFDAC2:538 | 4.95 |
| cfa-miR-196b | BAC1:10230 | BAC2:1931 | 2.41 |
| | BAC2:1931 | CFDAC2:4345 | 1.17 |
| | BAC1:10230 | CFDAC1:2254 | 2.18 |
| cfa-miR-592 | CFDAC1:2031 | CFDAC2:4893 | 1.27 |
| | BAC1:4316 | CFDAC1:2031 | 1.09 |
| cfa-miR-31 | BAC1:787 | BAC2:2290 | 1.54 |
| | BAC1:787 | CFDAC1:1890 | 1.26 |
| cfa-miR-138b | BAC1:1216 | BAC2:4243 | 1.8 |
| | BAC2:4243 | CFDAC2:985 | 2.11 |
| | BAC1:1216 | CFDAC1:599 | 1.02 |
| cfa-miR-490 | BAC1:1097 | BAC2:2406 | 1.13 |
| | BAC2:2406 | CFDAC2:487 | 2.31 |
| | BAC1:1097 | CFDAC1:493 | 1.15 |
| cfa-miR-496 | BAC1:978 | BAC2:2069 | 1.08 |
| | BAC2:2069 | CFDAC2:924 | 1.16 |
| cfa-miR-885 | BAC1:501 | BAC2:1161 | 1.21 |
| | BAC2:1161 | CFDAC2:538 | 1.11 |
| cfa-miR-196a | BAC1:858 | BAC2:84 | 3.35 |
| | CFDAC1:399 | CFDAC2:1340 | 1.74 |
| | BAC2:84 | CFDAC2:1340 | 3.99 |
| | BAC1:858 | CFDAC1:399 | 1.1 |

**Table 2** (*continued*)

| miRNA ID | Normalized reads | | \|log2(Fold_change)\| |
|---|---|---|---|
| | Sample 1 | Sample 2 | |
| cfa-miR-217 | CFDAC1:963 | CFDAC2:355 | 1.44 |
| | BAC1:334 | CFDAC1:963 | 1.53 |
| cfa-miR-493 | BAC1:405 | BAC2:834 | 1.04 |
| | BAC2:834 | CFDAC2:294 | 1.5 |
| | BAC1:405 | CFDAC1:188 | 1.11 |
| cfa-miR-889 | BAC1:502707 | CFDAC1:231678 | 1.12 |
| | BAC2:677126 | CFDAC2:285996 | 1.24 |

**Notes.**

BAC1, beagle adrenal cortex with treatment; BAC2, beagle adrenal cortex without treatment; BP1, beagle pituitary with treatment; BP2, beagle pituitary without treatment; CFDAC1, Chinese Field dog adrenal cortex with treatment; CFDAC2, Chinese Field dog adrenal cortex without treatment; CFDP1, Chinese Field dog pituitary with treatment; CFDP2, Chinese Field dog pituitary without treatment.

cfa-miR-125b) were found to be the most abundant known miRNAs in pituitary samples (The top 20 known miRNAs of abundance in pituitary were listed in Table S4). In previous high-throughput sequencing studies of other species, various isoforms of the miRNAs were always detected (*Li et al., 2011*; *Visser et al., 2014*). In this study, we found almost all known cfa-miRNAs were of isoforms, with a larger number of isoforms possessed by the miRNAs of higher abundance. Moreover, most of the isoforms showing the highest expression level were identical to the canonical forms in miRBase 21.0, but for some other miRNAs, like the cfa-miR-217, the isoforms that showed the highest expression level were not the canonical ones at all (Fig. S2), indicating the variability of miRNA expression pattern.

## Identification of novel miRNAs

In this study, a total of 147 novel miRNAs were identified across eight samples (Table S5). Compared with the known miRNAs identified here, rather lower expression levels were showed by the identified novel miRNAs. Of the 147 novel miRNAs, there were just 18 that were greater than 1,000 reads in expression level (Table S6). MiR-24_20231 and miR-21_17074 as the top two of the 18 novel miRNAs above, their predicted secondary structures were presented in Fig. S3. Of the 147 novel miRNAs, 75 shared a seed sequence with other known miRNAs of other species (especially mammals).

## Identification of differentially expressed known miRNAs

In this study, a total of 27 (6 up, 21 down), 37 (18 up, 19 down), 30 (22 up, 8 down), 21 (15 up, 6 down), 25 (19 up, 6 down), 58 (13 up, 45 down), 32 (7 up, 25 down) and 43 (32 up, 11 down) DE-miRNAs ($P\text{-value} < 0.001$, $\log_2(\text{Fold\_change}) > 1$) were detected in the comparisons of BAC1_vs_BAC2, BAC1_vs_CFDAC1, BAC2_vs_CFDAC2, CFDAC1_vs_CFDAC2, BP1_vs_BP2, BP1_vs_CFDP1, BP2_vs_CFDP2 and CFDP1_vs_CFDP2 respectively (the DE-miRNAs identified from adrenal cortex were listed in Table S7, and that from pituitary were listed in Table S8). Furthermore, a total of 90 miRNAs expressing differentially in pituitary comparisons were obtained, with 51 common DE-miRNAs shared in at least 2 pituitary comparisons (common DE-miRNAs). Of the 51 common DE-miRNAs, cfa-miR-105a, cfa-miR-219-3p and cfa-miR-802 were the

**Table 3** **The potential differentially expressed miRNAs targeting the differentially expressed genes, without prejudice to a negative correlation between miRNA and its corresponding target genes in both breeds.** MiRNA and its corresponding potential target genes' expression level are indicated by normalized reads and FPKM (Fragments Per Kilobase of exon model per Million mapped reads) respectively.

| miRNA & genes | Counts | | | |
|---|---|---|---|---|
| | BPC1 | BPC2 | TPC1 | TPC2 |
| **cfa-miR-135b** | 28942.56 | 15071.53 | 10065.27 | 21000.87 |
| *GRIN2B* | 0.250209 | 0.327438 | 4.27929 | 0.252423 |
| **cfa-miR-205** | 10730.32 | 22724.29 | 12764.71 | 55847.22 |
| *CHN1* | 11.6623 | 8.20414 | 58.7752 | 9.90749 |
| *GLRB* | 5.47912 | 5.66417 | 30.3379 | 6.2008 |
| *MMD* | 14.1617 | 12.4959 | 41.8962 | 12.9108 |
| **cfa-miR-30a** | 6285675 | 4436273 | 2553208 | 5289957 |
| *SLC1A2* | 3.18748 | 4.03293 | 108.28 | 2.35374 |
| *GRIN2A* | 0.0143535 | 0.0892129 | 1.75419 | 0 |
| *GRIA2* | 26.2931 | 26.7041 | 69.6217 | 19.3989 |
| *CAMTA1* | 4.61618 | 5.50413 | 14.7361 | 5.71324 |
| *SORCS3* | 12.1116 | 23.2745 | 12.4343 | 5.26901 |
| *ATP2B2* | 2.74647 | 3.10767 | 18.8807 | 3.18689 |
| *FAM49A* | 7.91528 | 8.88205 | 39.4249 | 8.66222 |
| **cfa-miR-34b** | 1331.3 | 1906.69 | 115.3 | 269.24 |
| *NRXN2* | 3.15428 | 1.36425 | 24.2216 | 2.60845 |
| *SCN2B* | 2.91752 | 1.15905 | 42.2646 | 1.72229 |
| *SLC6A1* | 7.34403 | 2.86335 | 32.8953 | 6.54531 |
| *SYNPO* | 4.77262 | 4.39135 | 21.2311 | 5.37488 |
| *GRM7* | 0.32311 | 0.330141 | 4.90747 | 0.391847 |
| *JAKMIP1* | 0.656131 | 0.477843 | 8.33219 | 0.783011 |
| **cfa-miR-802** | 812.1 | 346.67 | 115.3 | 1085.75 |
| *NECAB1* | 6.73288 | 4.95793 | 42.7074 | 7.71294 |

**Notes.**
BP1, beagle pituitary with treatment; BP2, beagle pituitary without treatment; CFDP1, Chinese Field dog pituitary with treatment; CFDP2, Chinese Field dog pituitary without treatment.

common DE-miRNAs of the four pituitary comparisons. Meanwhile, of the 62 DE-miRNAs detected in adrenal cortex comparisons, there were 39 common DE-miRNAs, with the three common miRNAs of cfa-miR-196a, cfa-miR-216b, and cfa-miR-514 shared in all these four adrenal cortex comparisons. In consideration of the functional impact exerted by miRNA expression level, we chose the DE-miRNAs whose total reads within pituitary or adrenal cortex were greater than 1,000 for subsequent analysis, thus resulting in a total of 39 and 18 common DE-miRNAs in pituitary and adrenal cortex separately (Fig. 3 and Table 2).

To validate the miRNA expression levels obtained by high-sequencing, 4 random DE-miRNAs, including cfa-miR-30a, cfa-miR-124, cfa-miR-205 and cfa-miR-222 were selected to perform qPCR (Fig. 4). The results showed cfa-miR-30a, cfa-miR-205 both were significantly downregulated in CFDP1_vs_CFDP2 ($|Log_2(fold change)| > 1$).
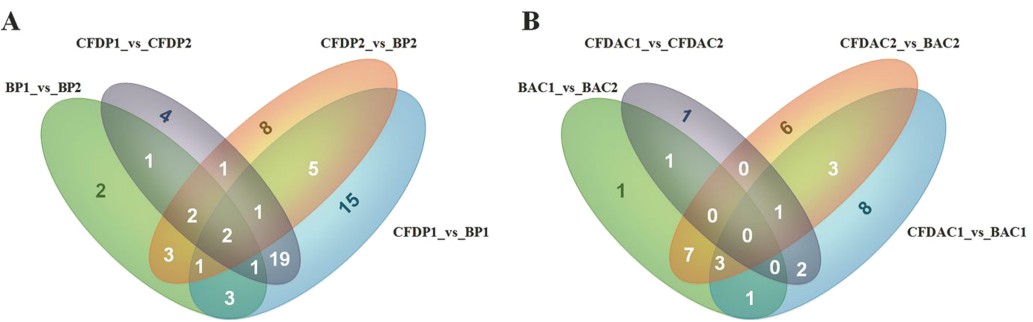

**Figure 3** **Venn diagrams of differentially expressed miRNAs (DE-miRNAs) among contrasts of each tissue.** (A) The DE-miRNAs distributed in pituitary, including BP1, BP2, CFDP1, and CFDP2. (B) The DE-miRNAs distributed in adrenal cortex, including four groups of BAC1, BAC2, CFDAC1, and CF-DAC2. The white number denotes the number of DEGs shared by at least two contrasts and were selected for further analysis.

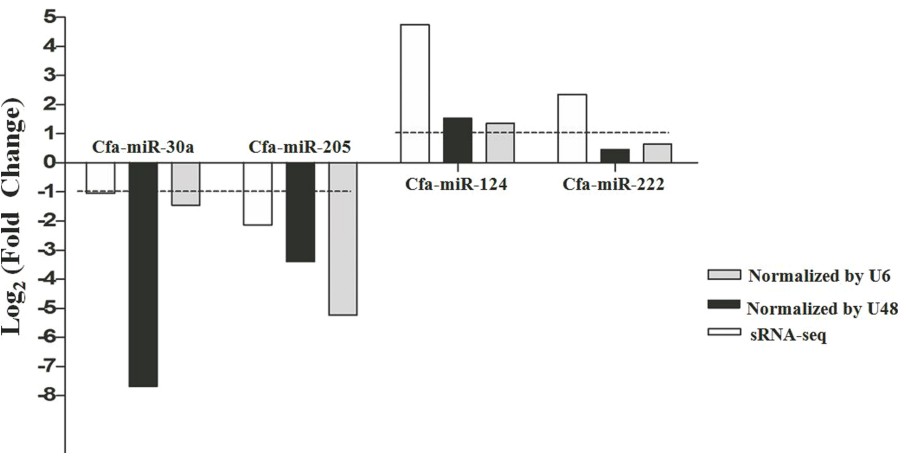

**Figure 4** **Validations of sRNA-seq by qPCR.** *U6* and *U48* are the internal control genes to normalize the expression level of cfa-miR-30a, cfa-miR-205, cfa-miR-124 and cfa-miR-222 in CFDP1_vs_CFDP2. CFDP1, CFD with chronic stress exposure; CFDP2, CFD without chronic stress exposure.

Cfa-miR-124 was significantly upregulated but not significantly for cfa-miR-222 in CFDP1_vs_CFDP2 ($|\text{Log}_2(\text{fold change})| > 1$). These observations were basically consistent with those obtained by high-sequencing, indicating the high-throughput sequencing data was reliable.

## miRNA target prediction, and GO and KEGG analysis

The common DE-miRNAs, including 33 and 22 in pituitary and adrenal cortex were used for miRNA target prediction, with the methods of Targetscan used. Consequently, 3,959 and 4,010 target genes were obtained from pituitary and adrenal cortex respectively (Table S9). Then 3,959 and 4,010 target genes were performed with GO and KEGG analysis

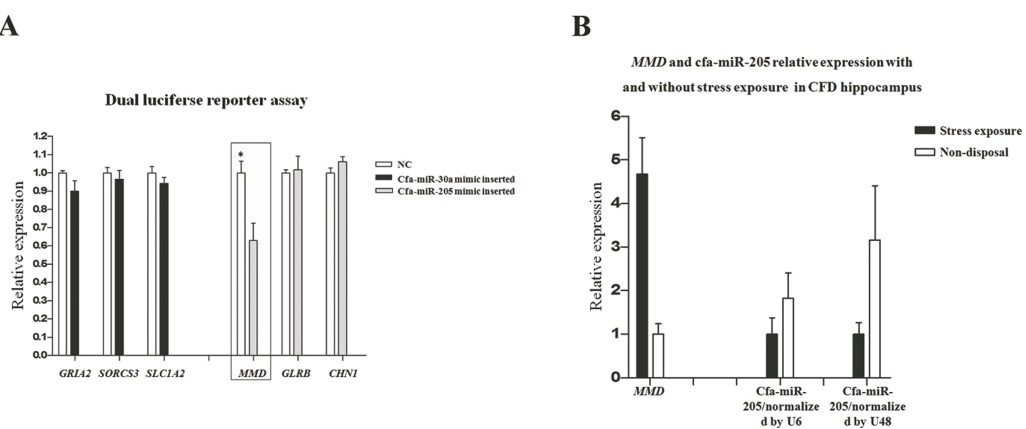

**Figure 5** (A) Cfa-miR-205-*MMD* pair identification. (B) Cfa-miR-205 and *MMD* expression patterns in CFD hippocampus with and without chronic stress exposure.

separately. Results showed that neither of the target genes from pituitary and adrenal cortex could be clustered into a certain GO term or KEGG pathway, due to the corrected $p$-value $> 0.05$. Instead, IPA was employed to identify the canonical pathways enriched by the predicted target genes. In CFDP1_vs_CFDP2 (i.e., 3,959 predicted target genes), the most enriched pathway was the axon guidance signaling pathway ($p = 1.34E-11$).

## DE-(miRNA-mRNA) pairs identification

In our previous transcriptome study (*Luo et al., 2015*), a total of 511 and 171 DEGs were identified in pituitary and adrenal cortex, respectively (Table S1). The DEGs identified in our transcriptome study were separately compared with the target genes in pituitary and adrenal cortex (i.e., 3,959 and 4,010 respectively), thus resulting in 105 and 10 common genes that were both included in DEGs and target genes (DE-target genes), respectively. Furthermore, the expression patterns of the 105 and 10 DE-target genes were compared with that of their corresponding miRNAs, without prejudice to the negative correlations between miRNAs and their target genes in both breeds. Consequently, 18 DE-(miRNA-mRNA) pairs in pituitary were obtained (Table 3). In consideration of the miRNA expression level and its potential target genes' relatedness to stress regulation, we selected 6 DE-(miRNA-mRNA) pairs to perform dual luciferase reporter assays to elucidate their dependability, including cfa-miR-30a-*GRIA2*, cfa-miR-30a-*SORCS3*, cfa-miR-30a-*SLC1A2*, cfa-miR-205-*MMD*, cfa-miR-205-*GLRB* and cfa-miR-205-*CHN1*. The dual luciferase reporter assay results showed just the cfa-miR-205-*MMD* conformed with the predicted (Fig. 5A), suggesting an important role played by cfa-miR-205 in regulating *MMD* under stress exposure in pituitary. Further, *MMD* and cfa-miR-205 expression levels were examined in the CFD brain, cerebellum, hippocampus and hypothalamus tissues by qPCR, normalized with *GAPDH*, *U6* and *U48* respectively. As expected, *MMD* was found to be upregulated, while downregulated for cfa-miR-205 in CFDP1_vs_CFDP2 (Fig. 5B).

## DISCUSSION

Dogs, especially beagles, are commonly used as medical experimental animals (*Lu et al., 2015*; *Ji et al., 2015*; *Li et al., 2015b*). In this study, we took advantage of the good manageability and good stress tolerance in beagles, and the excitability in CFD, to explore the differentially expressed miRNAs in pituitary–adrenal axis under chronic stress exposure within and between breeds, with miRNA-seq. Our miRNA-seq study yielded 28,667,214, 15,926,611, 32,583,375, 32,665,593, 27,591,701, 18,359,529, 81,028,035, and 51,202,889 raw reads in BAC2, BAC1, CFDAC2, CFDAC1, BP2, BP1, CFDP2 and CFDP1, respectively. To our knowledge, this is the first time that the miRNA profiles of dog pituitary and dog adrenal cortex were presented. We found cfa-miR-30 family (including cfa-miR-30a and cfa-miR-30d), cfa-miR-7 family (including cfa-miR-7 and cfa-miR-7g) and cfa-miR-125 (including cfa-miR-125a and cfa-miR-125b) were the miRNAs of highest activity in dog pituitary. Besides, cfa-miR-99a, cfa-miR-30 family (including cfa-miR-30a and cfa-miR-30d), cfa-miR-26 family (including cfa-miR-26a and cfa-miR-26b) and cfa-miR-7 family (including cfa-miR-7 and cfa-miR-7g) were the miRNAs of highest activity in dog adrenal cortex. These observations in pituitary and adrenal cortex were distinct from those in dog lung and dog trachea, in which the cfa-miRNA-143 and the cfa-let-7 were the ones of highest activity (*Zhao et al., 2014*) respectively. Furthermore, when we compared the miRNA expression profiles of pre- and post-stress exposure groups with that of their corresponding gene expression profiles, especially in pituitary, a strikingly converse expression pattern was observed between the miRNA expression profiles and gene expression profiles, indirectly proving that the negatively regulated pattern between miRNAs and genes existed. Intriguingly, in non-disposal groups, within both breeds, the total miRNA expression levels of the pituitary were significantly higher than that of the adrenal cortex (The total reads of 21, 22, and 23 nt detected in CFD pituitary were all greater than 15 million, and less than 8 million in CFD adrenal cortex, and this phenomenon was also found in beagles but not so significantly compared with CFD). Apart from the deviation caused by high-sequencing, this might be attributable to the wider roles played in biological functions by pituitary than that by adrenal cortex, as a larger miRNA reserve pool could irrigate more. Besides, in non-disposal groups, CFD pituitary exhibited a much higher miRNA expression level than that of beagles'. This might in part explain the different stress tolerance between beagles and CFD: Because the expression pattern of the miRNA was negatively correlated with its target genes, and the genes involving chronic stress response in pituitary were mainly upregulated, hence a larger miRNA reserve pool harbored by CFD pituitary indicated that the CFD had greater potential energy to upregulate the genes involving stress response to a greater degree, thus expressing symptoms more severely in CFD.

Because the miRNAs function as gene regulators, it's of limited significance to just discuss the roles of miRNAs without concerning their target gene expressions. In this study, we combined the miRNA expression profiles with their corresponding transcriptome profiles to analyze the potential DE-(miRNAs-target genes). Combined with the expression level and the relatedness to stress regulation, six DE-(miRNA-mRNA) pairs, including

cfa-miR-30a-*GRIA2*, cfa-miR-30a-*SORCS3*, cfa-miR-30a-*SLC1A2*, cfa-miR-205-*MMD*, cfa-miR-205-*GLRB* and cfa-miR-205-*CHN1* were identified as the pivotal candidate miRNAs and their corresponding target genes. The further dual luciferase reporter assay indicated a significant target relation between cfa-miR-205 and *MMD*. We identified one pivotal DE-(miRNA-mRNA) pair, and this could be partly accounted for our rigorous strategy of the statistical analysis and the control group set, thus inducing some true positive DE-(miRNA-mRNA) pairs being omitted. More importantly, these results strongly indicated an actual positive correlated expression pattern between most miRNAs and their corresponding genes under chronic stress exposure, indicating a general buffering role for miRNAs in gene expression but not a role of synergy. In this study, miRNAs that were differentially expressed in at least two comparisons within pituitary or adrenal cortex groups were regarded as the candidate miRNAs for target prediction, and the target genes expressing differentially were selected. In addition, we didn't take the miRNA and its corresponding target genes into consideration if the expression pattern between the miRNA and its target genes was just negatively correlated in one breed but not in another. These measures all contributed to a smaller number of the candidate DE-miRNAs and their corresponding target genes, but helped a lot in identifying the miRNAs of major effect on chronic stress regulation.

In this study, through conjoint analysis of mRNA-seq and miRNA-seq studies, we identified cfa-miR-205 might play a important role in regulating *MMD* in pituitary under chronic stress exposure. *MMD* (monocyte to macrophage differentiated-associated) has been found to be up-regulated upon monocyte differentiation (*Liu et al., 2012*). In our previous hippocampal sections upon the CFD and the beagle with and without chronic stress exposure, we found the CFD hippocampus underwent stress exposure and exhibited the largest apoptosis compared with other groups (*Luo et al., 2015*), suggesting a potential immune response in it. Thus, in CFDP1_vs_CFDP2, we hypothesized *MMD* should be upregulated, along with the cfa-miR-205 be down-regulated, as more macrophages were transferred from monocytes as supposed. Inspiringly, the qPCR results supported our hypothesis (Fig. 5B), indicating a potential pivotal role for cfa-miR-205 in regulating *MMD* under chronic stress exposure.

## CONCLUSIONS

In conclusion, we compared the miRNA expression profiles of the adrenal cortex and pituitary in beagles and CFD with and without stress exposure. We detected 425 known miRNAs and 147 novel miRNAs in total across 8 samples, and of the 425 known miRNAs, a total of 90 and 62 miRNAs were found to be differentially expressed in pituitary and adrenal cortex respectively. In combination with the transcriptome profiles corresponding to the miRNA profiles of this study, we found cfa-miR-205 might play an important role by upregulating *MMD*, thus enhancing the immune response in the pituitary and hippocampus, under chronic stress exposure. Our results shed light on the miRNA expression profiles in pituitary and adrenal cortex with and without chronic stress exposure, and provide a new insight into miR-205 with its feasible role in regulating chronic stress in the pituitary and hippocampus through targeting *MMD*.

### Funding

This research was supported by the National Natural Science Foundation of China (31101677). The funders had no role in study design, data collection and analysis, decision to publish, or preparation of the manuscript.

### Grant Disclosures

The following grant information was disclosed by the authors:
National Natural Science Foundation of China: 31101677.

### Competing Interests

The authors declare there are no competing interests.

### Author Contributions

- Wei Luo performed the experiments, analyzed the data, wrote the paper, prepared figures and/or tables.
- Meixia Fang and Huijie Xing performed the experiments, contributed reagents/materials/analysis tools.
- Haiping Xu reviewed drafts of the paper.
- Jiangnan Fu and Qinghua Nie conceived and designed the experiments.

### Animal Ethics

The following information was supplied relating to ethical approvals (i.e., approving body and any reference numbers):

In this experiment, all animals used were approved by the Animal Care Committee of Jinan University (Guangzhou, People's Republic of China), approval number 20131018001 and implemented in accordance with the experimental basic principles.

### Data Availability

NCBI GEO Database
GSE72015
http://www.ncbi.nlm.nih.gov/geo/query/acc.cgi?acc=GSE72015.

### Supplemental Information

Supplemental information for this article can be found online at http://dx.doi.org/10.7717/peerj.1682#supplemental-information.

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
