# Peer review of "Comparison of miRNA expression profiles in pituitary–adrenal axis between Beagle and Chinese Field dogs after chronic stress exposure"

_PeerJ, doi:10.7717/peerj.1682_

## Round 0.1 · original submission · Major Revisions

Dear Drs. Fu and Nie,

Your manuscript entitled "Comparison of miRNA expression profiles in pituitary–adrenal axis between Beagle and Chinese Field dogs after chronic stress exposure" has been reviewed. The reviewers have recommended your manuscript for major revisions before it can be considered further for publication. Therefore, I invite you to respond to the reviewer(s)’ comments and revise your manuscript.

I do apologize for the delay of review process while we are waiting for the reviews comments.

Once again, thank you for submitting your manuscript to the PeerJ and I look forward to receiving your revision.

Sincerely,
Dr. Cong-jun Li

Reviewer 1 ·

Basic reporting

No Comments

Experimental design

No Comments

Validity of the findings

1. The authors used only one housekeeping gene (U6) in the QRT-PCR assays which were proved to need at least three housekeeping genes to ensure the testing accuracy. Please add other two internal control genes in the QRT-PCR assay.
2. As the important conclusion of this study, the targeting relationship between Cfa-miR-30a and candidate genes (SLC1A2, GRIA2, GRIN2A and SORCS3) was not validated by experimental methods (dual-luciferase reporter system). Please improve this part.

Additional comments

Luo et al. conducted a meaningful study to investigate the miRNAs in pituitary and adrenal cortex that regulated chronic stress response with high-throughput sequencing in Beagle and Chinese Field Dogs, and find that cfa-miR-30a possibly functionally targeted SLC1A2, GRIA2, GRIN2A, and SORCS3, thereby affecting the neuronal excitability, the synaptic plasticity and the neuronal activity in chronic stress. However, there are some serious issues in the current study.
1. Why the authors did not use the latest version of miRbase (miRbase 21.0) in the identification of known miRNAs? Please improved this part.
2. Why the authors uses RNA pools to construct the small RNA-seq libraries? This operation will directly affect the accuracy of identification of DE-miRNAs, due to the lack of biological repeats. Please improve or clarify this part.
3. In target prediction of DE miRNAs in this study, the authors used miRD and TargetScan softwares based on mouse or human genes, which were quite distinct with dog genes in 3’ UTR sequences. Why the authors did not use their RNA-seq data (porcine 3’ UTR sequences) to complete the target prediction to get the highest accuracy? Please improve this part.
4. The authors used only one housekeeping gene (U6) in the QRT-PCR assays which were proved to need at least three housekeeping genes to ensure the testing accuracy. Please add other two internal control genes in the QRT-PCR assay.
5. Please provide the accession number of the raw small RNA-seq Data in the revised manuscript.
6. As the important conclusion of this study, the targeting relationship between Cfa-miR-30a and candidate genes (SLC1A2, GRIA2, GRIN2A and SORCS3) was not validated by experimental methods (dual-luciferase reporter system). Please improve this part.

Reviewer 2 ·

Basic reporting

In the manuscript “comparison of miRNA expression profiles in pituitary-adrenal axis between beagle and Chinese Field dogs after chronic stress exposure”, the authors compared the expression profiles of miRNAs in pituitary and adrenal cortex tissues with or without stress stimulation. Some miRNAs associated with stress response have been identified. The functions of miR-30 and its target genes in the stress response have also been analyzed. This study offered useful information about miRNA in the stress response in dogs.

Questions and suggestions
1. The abundance miRNAs have been listed in Table 2 and 3, while the differential expressed miRNAs between stimulation and control groups have not and neither the differential express miRNAs between two strains. I think the later part is more important, I would suggest that the author list these miRNAs in tables.
2. In the discussion part, the authors concluded that the total miRNA expression in no-disposal groups were higher than that of control group based on fig.1. I think this conclusion may have some problems. For me, I think the difference of the total reads of miRNAs of 20~22nt was just caused by the depth of sequence. In addition, based on the distribution, the 31-32nt RNAs seem more interesting. The proportions of these longer RNAs were significant increased after stress stimulation.
3. The authors finally focused on the miR-30. Are there any other important miRNAs or other pathways involved into stress responses? I would suggest the author further analyze the data. Also, there were several target genes of miR-30 have been predicted. It would be better to confirm these target genes using LUC assay or Western blotting.

Experimental design

No Comments

Validity of the findings

No Comments

Additional comments

No Comments

Reviewer 3 ·

Basic reporting

This manuscript have investigated the miRNA expression profiles of the adrenal cortex and pituitary in two dog breeds Beagle and CFD with and without stress exposure. This study is well designed and written with a good English.

Experimental design

The design of this study is solid with robust statistical methods.

Validity of the findings

The design of this study is solid with robust statistical methods.

Additional comments

This manuscript have investigated the miRNA expression profiles of the adrenal cortex and pituitary in two dog breeds Beagle and CFD with and without stress exposure. This study is well designed and written with a good English.
Some questions and coments may need to be clarified and considered before publication. As is shown in Table 1. Statistics of miRNA-seq across 8 samples, we can find a large range the number the raw reads and clean read? how to consider the bias caused by the unbalanced panel data in the DE analysis ? In the "2.5 Sequence data analysis", the authors need to clarify how to map the clean read? which methods or tools were used? and which dog genome assembly, "canFam x" used in this stduy.? In Line 169, the manuscript need to clarify the statiscticss procedures for estimate the DE-miRNAs, how to mesure the fold change? In Line 174-176, the author should descript the rules used for target prediction, not just list the softwares.

Some minor comments as follows,
1.Line 137 "2.4 small RNA library construction and sequencing" , capital "S"
2.Line 176, delete " it’s just the ".
3.Page 26, " Distribution ration by miRNAs on genome" should be "ratio
4.In Fiugre 2, what is the chrM mean? Have you consider the sex chromosome ?

---

## Round 0.2 · accepted · Accept

Congratulations on the Acceptance

Reviewer 1 ·

Basic reporting

No comments

Experimental design

No comments

Validity of the findings

No comments

Additional comments

In the new version, the authors have improved all the areas I commentted in previous version of manuscript, and the new manuscript is well written, and there is no problem in the statistical method, presenting results, discussion and references.

Reviewer 3 ·

Basic reporting

The authors have responded in an appropriate manner to the indications of the review, and I have no additional comments.

Experimental design

The authors have responded in an appropriate manner to the indications of the review, and I have no additional comments.

Validity of the findings

The authors have responded in an appropriate manner to the indications of the review, and I have no additional comments.

Additional comments

The authors have responded in an appropriate manner to the indications of the review, and I have no additional comments.